# Laxative Effects of Phlorotannins Derived from *Ecklonia cava* on Loperamide-Induced Constipation in SD Rats

**DOI:** 10.3390/molecules26237209

**Published:** 2021-11-28

**Authors:** Ji-Eun Kim, Yun-Ju Choi, Su-Jin Lee, Jeong-Eun Gong, You-Jung Jin, So-Hae Park, Hee-Seob Lee, Young-Whan Choi, Jin-Tae Hong, Dae-Youn Hwang

**Affiliations:** 1Laboratory Animal Resources Center, Department of Biomaterials Science (BK21 FOUR Program), Life and Industry Convergence Research Institute, College of Natural Resources & Life Science, Pusan National University, Miryang 50463, Korea; prettyjiunx@naver.com (J.-E.K.); poiu335@naver.com (Y.-J.C.); nuit4510@naver.com (S.-J.L.); jegog@naver.com (J.-E.G.); hjinyuu1@naver.com (Y.-J.J.); sohaehw@pusan.ac.kr (S.-H.P.); 2Department of Food Science and Nutrition, College of Human Ecology, Pusan National University, Busan 46241, Korea; heeseoblee@pusan.ac.kr; 3Department of Horticultural Bioscience, Life and Industry Convergence Research Institute, College of Natural Resources & Life Science, Pusan National University, Miryang 50463, Korea; ywchoi@pusan.ac.kr; 4College of Pharmacy, Chungbuk National University, Chungju 28160, Korea; jinthong@chungbuk.ac.kr

**Keywords:** constipation, phlorotannins, laxative effects, muscarinic acetylcholine receptors

## Abstract

This study investigated the laxative effects of phlorotannins (Pt) derived from *Ecklonia cava* (*E. cave*) on chronic constipation by evaluating alterations in stool parameters, gastrointestinal motility, histopathological structure, mucin secretion, gastrointestinal hormones, muscarinic cholinergic regulation, and fecal microbiota in SD rats with loperamide (Lop)-induced constipation subjected to Pt treatment. Stool-related parameters (including stool number, weight, and water contents), gastrointestinal motility, and length of intestine were significantly enhanced in the Lop+Pt-treated group as compared to the Lop+Vehicle-treated group. A similar recovery was detected in the histopathological and cytological structure of the mid-colon of Lop+Pt-treated rats, although the level of mucin secretion remained constant. Moreover, rats with Lop-induced constipation subjected to Pt treatment showed significant improvements in water channel expression, gastrointestinal hormone secretions, and expression of muscarinic acetylcholine receptors M2/M3 (mAChRs M2/M3) and their mediators of muscarinic cholinergic regulation. Furthermore, the Lop+Pt-treated group showed a significant recovery of *Bifidobacteriaceae*, *Muribaculaceae*, *Clostridiaceae*, and *Eubacteriaceae* families in fecal microbiota. Taken together, these results provide the first evidence that exposure of SD rats with Lop-induced constipation to Pt improves the constipation phenotype through the regulation of membrane water channel expression, GI hormones, the mAChR signaling pathway, and fecal microbiota.

## 1. Introduction

The marine plant *E. cava* has long been reported to have various therapeutic effects, including anti-aging [1], antioxidation [2], anti-inflammation [3], anticancer [4], neuroprotection [5], antiplasmin activity [6], and tyrosinase inhibitory activity [7]. These functions are closely linked to the constituent components, such as carotenoids, fucoidan, sulfated polysaccharides, peptides, and tannins [8,9,10]. Of these, tannins have received great attention due to their multiple biological functions, including antioxidative, cardioprotective, antitumor, antibacterial, antiviral, anti-inflammatory, antimutagenic, and immune-modulatory effects [11,12]. Depending on the base unit or monomer, they are classified as phlorotannins (Pt), hydrolyzable tannins (HT), and condensed tannins (CT) [12]. Pt consists of a phloroglucinol unit with various degrees of polymerization and is produced as a secondary metabolite from marine brown algae such as *Sargassum fusiforme*, *Fucus vesiculosus*, *Alaria esculenta*, *Laminaria digitata*, and *Ascophyllum nodosum* [13,14,15].

Pt is further classified into four subclasses: fuhalols and phlorethols, fucols, fucophlorethols, and eckols, although numerous types of Pt are detected in brown algae [16,17]. They possess significantly high biological activities, including antioxidant [2,18], human immunodeficiency virus type-1 (HIV-1) inhibitory [19], anti-inflammatory [20], radioprotective [21], antidiabetic [22], and antiproliferative effects [23]. However, the laxative effects of Pt on constipation have not been investigated in in vitro and in vivo systems. Previous studies on the laxative effects of natural products containing several classes of tannins have provided strong evidence of the potential laxative effects of Pt. An extract of *Aloe ferox* Mill. containing phenol and tannin significantly reduced the symptoms of constipation, including stool number and weight, as well as the gastrointestinal transit ratio, in a Lop-treated animal model [24]. Other studies have reported the recovery of several stool markers, water content, histological analysis, and gastrointestinal (GI) hormone concentrations in a model of Lop-induced constipation after exposure to *Liriope platyphylla* aqueous extract containing 16.75% tannins [25,26]. Moreover, *E. cava*, the raw material of phlorotannin, induced the recovery of the histopathological structure, mucin secretion, and gastrointestinal hormone secretion in a Lop-induced constipation model [27].

The current study, therefore, aimed to investigate the laxative effects and mechanism of Pt on Lop-induced constipation in SD rats. The data presented here constitute strong evidence that Pt is a powerful candidate for alleviating the symptoms of constipation through the regulation of water content, muscle contraction, and fecal microbial composition.

## 2. Results

### 2.1. Confirmation of Pt Purified from E. cava

We first confirmed the composition of Pt purified from *E. cava*. LC–MS analysis contained several compounds, including phloroglucinol, phlorofucofuroeckol A, fucofuroeckol, and fucodiphloroethol G as the main substances (Figure 1a,b). These data indicate that Pt was successfully extracted from *E. cava*.

### 2.2. Effect of Pt Administration on Feeding Behavior and Excretion Parameters

We first investigated whether Pt administration stimulates defecation in constipated rats. To achieve this, alterations in food intake, water consumption, and stool parameters were analyzed in constipated models after administration of Pt. The body weight and water consumption remained at a constant level in all three excrement groups, although decreased food intake was observed in the Lop+Vehicle- and Lop+Pt-treated groups, as compared to the non-treated (No) group. However, compared to the Lop+Vehicle-treated group, urine volume was significantly increased in the Lop+Pt-treated group (Figure 1c). Moreover, the number, weight, and water content of stools were markedly increased in the Lop-induced constipation models after administration of Pt. The stool morphology sufficiently reflected the above alterations of stool parameters (Figure 1d). These data show that administration of Pt promotes defecation and urination in the Lop-induced constipation model in SD rats, without significant alterations in their food intake and water consumption.

### 2.3. Effect of Pt Administration on GI Motility and Intestinal Length

Next, we examined whether the stimulatory effects of Pt on defecation are accompanied by alterations in the motility function of the intestine. The charcoal meal transit test and intestine length were analyzed in SD rats with Lop-induced constipation after Pt administration. The propulsion of charcoal meal was 24% lower in the Lop+Vehicle-treated group than the non-treated group. However, these changes markedly recovered after Pt administration (Figure 2). In addition, a similar recovery pattern was detected for the intestine length, although the recovery rate varied (Figure 2). These results indicate that the stimulatory effects of Pt on defecation are associated with the regulation of GI motility and intestinal length in SD rats with Lop-induced constipation.

### 2.4. Effect of Pt Administration on Histopathological and Cytological Structure of the Mid-Colon

To investigate changes in the histological and cytological structure of the mid-colon due to the defecation stimulation effects of Pt in constipated rats, we examined alterations in the histological and cytological structures in mid-colon sections stained with H&E solution by analyzing them with TEM. In the H&E staining analysis, the lengths of crypts, the thickness of the flat luminal surface, and the muscle of the mid-colon were significantly decreased in the Lop-treated group relative to the non-treated group. However, these decreases were markedly enhanced in the Lop+Pt-treated group. A complete recovery was observed in the length of crypts and muscle thickness in the mid-colon sections of Lop+Pt-treated rats (Figure 3a). Similar alteration patterns were detected in the TEM analysis of the mid-colon. The decreased number of goblet cells after Lop treatment was significantly increased in the Lop+Pt-treated group, but the reverse pattern was observed for the number of lipid droplets (Figure 3b). Taken together, the above results indicate that enhanced stool excretion after Pt administration is completely reflected in the alterations observed in the histological and cytological structures of the mid-colon in the Lop-induced constipation SD rat model.

### 2.5. Effect of Pt Administration on Regulatory Function of Mucin and Water

To examine whether the defecation stimulation effects of Pt are accompanied by alterations in the regulation of mucin and water secretion, the levels of mucin secretion and water channel expression were analyzed in the mid-colons of non-treated, Lop+Vehicle-treated, and Lop+Pt-treated groups. Blue-stained mucin in the goblet cells of the crypt was significantly decreased in the Lop+Vehicle-treated group as compared to the non-treated group. Moreover, Pt administration did not induce the recovery of mucin secretion in the mid-colon of SD rats (Figure 4a). These changes were completely reflected in the transcription level of the MUC2 gene (Figure 4b). Furthermore, analysis of the transcription levels of water channel genes in the mid-colons of subset groups revealed markedly decreased levels of AQP3 and AQP8 mRNA in the Lop+Vehicle-treated group as compared to the non-treated group, which were determined to be completely recovered after Pt administration (Figure 4b). These results indicate that the defecation stimulation effects of Pt are associated with enhanced expression of the membrane water channel in the mid-colon of SD rats with Lop-induced constipation, without affecting the secretion of mucin.

### 2.6. Effect of Pt Administration on Regulation of GI Hormone Secretions

To examine the correlation between the defecation stimulation effects of Pt and the secretion levels of GI hormones, the concentrations of CCK, gastrin, and SS were measured in the mid-colon homogenate of Lop+Pt-treated SD rats. The levels of the three hormones showed similar regulation patterns in rats of all subset groups. The concentrations of the three GI hormones were decreased in the Lop+Vehicle-treated group compared with the non-treated group, with significantly increased concentrations in the Lop+Pt-treated group, although the increase rate varied (Figure 5). These results indicate that the defecation stimulation effects of Pt may be associated with stimulating the secretion of GI hormones in rats with Lop-induced constipation.

### 2.7. Effect of Pt Administration on the Downstream Signaling Pathway of mAChRs

Next, we investigated the correlation between the defecation stimulation effects of Pt and the mAChR signaling pathway. To achieve this, the expression levels of mAChR M2, mAChR M3, Gα, PKC, p-PKC, PI3K, and p-PI3K proteins were determined in the mid-colons of subset groups. The expression levels of mAChR M2 and mAChR M3 in the Lop+Vehicle-treated group were lower than the non-treated group but were enhanced in the Lop+Pt-treated group (Figure 6). Moreover, the reverse pattern was observed in the mAChR downstream signaling pathway. The phosphorylation levels of PKC, PI3K, and MLC were dramatically increased in the Lop+Vehicle-treated group as compared to the non-treated group but were observed to recover after Pt administration, although with varying recovery rates (Figure 6). A similar regulatory pattern was detected for Gα protein expression (Figure 6). These data indicate that the defecation stimulation effects of Pt may be associated with the recovery of mAChR expression and their signaling pathway in the mid-colon of constipated SD rats.

### 2.8. Effect of Pt Administration on the Profile of Fecal Microbiota

Finally, we investigated whether the defecation stimulation effects of Pt are associated with alterations in the fecal microbiota profile of rats with Lop deficiency-induced constipation. To achieve this, the overall microbial composition was analyzed in stool samples of non-treated, Lop+Vehicle-treated, and Lop+Pt-treated groups. In the principal component analysis plot, the fecal microbial community of Lop+Vehicle-treated mice was significantly different from the non-treated WT rats. However, these alterations showed no recovery after Pt treatment (Figure 7b). A similar pattern of colony formation on agarose plates was observed (Figure 7a). Analysis of the microbial phyla revealed an increased population of Candidatus Melainabacteria and Verrucomicrobia in the non-treated group as compared to the Lop+Vehicle-treated group, but a significant recovery was detected only for Candidatus Melainabacteria after Pt treatment, while the population of Verrucomicrobia was further increased after exposure (Figure 8a). Furthermore, a significant recovery at the class level was detected in only Actinomycetia, Clostridia, Negativicutes, and Deltaproteobacteria after exposure to Pt, although Lop treatment induced increases in the populations of Erysipelotrichia and Gammaproteobacteria (Figure 8b). Moreover, a significant increase or decrease at the family level was detected in Bifidobacteriaceae, Muribaculaceae, Prevotellaceae, Clostridiaceae, and Eubacteriaceae in the fecal samples of the Lop-treated group. However, pronounced recovery was observed for Bifidobacteriaceae, Muribaculaceae, Clostridiaceae, and Eubacteriaceae after treatment with Pt, although the recovery rates were different (Figure 8c). These results indicate that the defecation stimulation effects of Pt are closely associated with alterations in some populations of fecal microbiota in SD rats with Lop-induced constipation.

## 3. Discussion

Studies on the function and chemical properties of Pt have provided scientific evidence for its potential as a drug for treating constipation [28]. In particular, the extract of *E. cava*, a well-known major source of Pt, resulted in considerable improvement of chronic constipation in the Lop-induced SD rat model [27]. As part of our research to elucidate new functions of Pt, we investigated the laxative effects and mechanism of Pt in SD rats with Lop-induced constipation. Our results provide the first evidence that Pt treatment improves the symptoms of chronic constipation, including delayed stool excretion, decreased gastrointestinal transit, and abnormal histopathological and cytological structures in the Lop-induced constipation model. Moreover, our data generated in the Lop-induced constipation model revealed that the laxative effects of Pt are probably associated with the regulation of water channel expression, GI hormone concentrations, mAChR signaling pathway, and fecal microbiota.

Previous studies have reported that tannin-containing extracts exert beneficial effects that improve various constipation symptoms. Maesil (the fruit of *Prunus mume* Siebold & Zucc.) increased the frequency and moisture contents of feces as well as GI motility in a low-fiber-diet-induced constipation model [29]. Similar effects, including increased feces volume and improvement of intestinal motility, were observed in rats with Lop-induced constipation after treatment with *Morella serrata* extract (300 mg/kg) for 7 days, wherein the galla rhois administered contained high levels of tannins [30,31]. Furthermore, three brown algae (*E. cava*, *Laminaria japonica*, and *Undaria pinnatifida*, all containing Pt as the main source) have been observed to have laxative effects in animal models and humans. Some significant recoveries of stool parameters, GI transit, mucin secretion, GI hormone concentrations, and the mAChR signaling pathway were previously evaluated in a Lop-induced model treated with *E. cava* and *L. japonica* [27,32]. In 22 healthy individuals suffering from low defecation frequency, treatment with *U. pinnatifida* induced an increased frequency of defecation and alteration of intestinal microbiota [33]. In the current study, we evaluated the laxative effects of Pt derived from *E. cava* to verify whether the laxative effects of brown algae are related to Pt. Our results provide strong evidence that Pt has potential applications in the treatment of constipation.

Since the efficacy of *E. cava* has been well-documented in previous studies, it is necessary to directly compare the laxative effects of *E. cava* and Pt [27]. Among the various analytical factors, recoveries in the water contents of stools, histological structure of the colon, and the expression levels of AQPs were higher in the *E. cava*-treated group than the Pt-treated group. However, the reverse pattern was detected for other factors, including the concentrations of GI hormones and the mAChR signaling pathway, although the rates varied [27]. Notably, a dramatic difference was observed in the secretion and expression of mucin. A significant increase or recovery was detected in the *E. cava*-treated constipation model, whereas the Pt-treated constipation model remained at a constant level (Figure 4). These results indicate that the laxative effects of Pt may be closely linked to the regulation of GI hormones and the mAChR signaling pathway in the mid-colon of the Lop-induced constipation model.

GI hormones are composed of two structurally homologous families, and responses to GI functions include the contraction of smooth muscles, enzyme secretion, fluid secretion, tissue growth, and GI peptide secretion [34]. The first family group consists of gastrin and CCK, and they play a key role in the digestive mechanism through the regulation of gastric acid secretion, digestive enzyme secretion, and gut motility. The second family includes secretin and glucagon and affects various biological activities, including glucose metabolism, glycogen synthesis, and gluconeogenesis [35,36]. In particular, the laxative effects of some natural products on constipation are closely linked to hormones in the first group. Previous studies have reported that the concentrations of CCK and gastrin were significantly increased in the Lop-induced constipation model after exposure to spicatoside A as compared to the Lop+Vehicle-treated group [37]. Similar enhancement effects were detected in the galla rhois-treated constipation model, although the increase rate varied [38]. Furthermore, a tannin-enriched extract of *E. cava* induced an increase in CCK and gastrin concentrations in SD rats with loperamide-induced constipation [27]. In the current study, we evaluated the concentrations of CCK and gastrin to examine whether the defecation stimulation effects of Pt are accompanied by the regulation of GI hormones. We determined that the decreased levels of these hormones were recovered after Pt treatment. These results are in agreement with previous studies, in which the concentrations of CCK and gastrin were increased after exposure to spicatoside A, galla rhois, and *E. cava*. Thus, the results of the present study provide the first evidence that the defecation stimulation effects of Pt are strongly correlated with changes in CCK and gastrin concentrations in the Lop-induced constipation model.

AChRs are widely distributed in neuronal cells and non-neuronal cells in the gut and are classified into two groups: metabotropic G-protein-coupled muscarinic receptors and ionotropic nicotinic receptors [39]. Among these receptors, mAChRs play a role in the function of gut epithelial cells, which regulate food degradation and the uptake of water, nutrients, and electrolytes via binding to ACh molecules [39]. To date, some studies have reported that the laxative effects of natural products are linked to the mAChR signaling pathway in the Lop-induced constipation model. Uridine and spicatoside A were reported to induce the recovery of mAChR expression and their downstream signaling pathway in Lop-treated SD rats [37,40]. Similar recovery effects on the expression of mAChRs and their signaling pathways were detected in the constipation model after treatment with *L. platyphylla* and galla rhois [26,31]. In the current study, significant recovery of the expression of mAChRs and their signaling pathway was detected in the mid-colon of constipated rats treated with Pt, thereby indicating that the mAChR signaling pathway plays a key role in the laxative effects of Pt.

Pt compounds are produced as highly hydrophilic components through the polymerization of phloroglucinol (1,3,5-trihydroxybenzene) monomers and the acetate–malonate pathway [41]. They are classified into four subclasses: fuhalols and phlorethols, fucols, fucophloroethols, and eckols [9]. Notably, some subclasses of Pt from *E. cava* have been isolated and characterized. They include phloroglucinol [42], eckol [42,43], fucodiphloroethol G [42], phlorofucofuroeckol A [42,43], 7-phloroeckol [42], dieckol [42], and 6,6′-bieckol [18,42], although they have been reported in different papers. In the present study, LC–MS analyses identified only five compounds, namely, phloroglucinol, phlorofucofuroeckol A, fucofuroeckol, and fucodiphloroethol G. This difference is thought to be due to the extraction method and the quality of the raw material used.

## 4. Materials and Methods

### 4.1. Preparation of Pt from E. cava

Pt contents were extracted according to the methods described by Lee et al. [44], with slight modifications. Briefly, the first extract was collected from dried powder of *E. cava* (30 g) in 70% ethanol (300 mL; *v*/*v*) at 37 °C for 12 h using a shaking incubator (VS-8480; Vision Scientific, Bucheon, Korea; 200 rpm). After filtration with Whatman No. 2 filter paper (Whatman, Brentford, UK), the filtrate was evaporated at 40 °C using a rotatory vacuum evaporator (EYELA, N-1100 series, Tokyo, Japan). The ethanol extract was dissolved in water and further partitioned with n-hexane, chloroform, and ethyl acetate. The ethyl acetate fraction is known to be rich in Pt, as reported in previous studies [44,45]. The solvent was evaporated at 40 °C with subsequent freeze-drying of the sample and finally stored at −20 °C until required.

### 4.2. Liquid Chromatography–Mass Spectrometry (LC–MS) Analysis

LC–MS analysis was performed using the Agilent 1290 Infinity HPLC system (Agilent Technologies, Waldbronn, Germany), coupled with a hybrid quadrupole time-of-flight (Q-TOF) mass spectrometer (6530, Agilent Technologies). LC–MS signals were detected on a mass spectrometer operating in the negative ionization mode. An ACQUITY UPLC BEH C18 column (2.1 × 100 mm, 1.7 µm) (Waters, Milford, MA, USA) was applied for chromatographic separation under the following conditions: 0.3 mL/min flow rate, 10 µL injection volume, 0.1% formic acid–water as mobile phase A, and 100% acetonitrile as mobile phase B. The operating parameters applied for MS detection were as follows: gas temperature, 300 °C; gas flow, 9 L/min; nebulizer pressure, 45 psig; sheath temperature, 350 °C; sheath gas flow, 11 L/min; VCap, 4000 V; fragmentor voltage, 175 V. All acquisitions and data analysis were controlled by the MassHunter software (version B. 0600, Agilent Technologies). In this study, ultra-performance hybrid quadrupole time-of-flight (UPLC–Q-TOF-MS) methods were established for profiling the components of Pt compounds representing fucodiphloroethol G and phloroglucinol. Compounds representing 31.88 and 9.35% of the total peak area of fucodiphloroethol G and phloroglucinol were identified. Results from this analysis reveal that the Pt used in this study comprised several main ingredients, including phloroglucinol, phlorofucofuroeckol A, fucofuroeckol, docosanoic acid, and fucodiphloroethol G (Figure 1b).

### 4.3. Experimental Design for Animal Study

The Pusan National University Institutional Animal Care and Use Committee (PNU-IACUC) reviewed and approved the protocol for animal experiments (approval number PNU-2019-2458). Eight-week-old Sprague Dawley (SD) rats used to study the therapeutic effects of Pt were provided by Samtako BioKorea Inc. (Osan, Korea) and were allowed to adapt for 1 week. All animals were bred at the Pusan National University Laboratory Animal Resources Center, accredited by the Korea Food and Drug Administration (KFDA) (accredited unit number 000231) and the Association for Assessment and Accreditation of Laboratory Animal Care (AAALAC) International (Accredited Unit Number 001525) throughout the experimental period. These facilities were maintained in a specific pathogen-free state (SPF) under a strict light cycle (lights on at 08:00 h and off at 20:00 h) at 22 ± 2 °C and relative humidity 50 ± 10%. SD rats were provided *ad libitum* access to standard irradiated chow diet (Samtako BioKorea Co., Osan, Korea).

All animal laxative effects of Pt were measured as described in previous studies [31,46]. Briefly, all rats (male, 270 g, *n* = 30) were allocated to one of three groups, including the non-constipation group (No group, *n* = 10) or a constipation group (*n* = 20). Constipation in rats was induced by subcutaneously injecting Lop (4 mg/kg weight) in 0.5% Tween 20 in 1 × PBS twice daily (9 a.m. and 6 p.m.) for 3 days. At 9 a.m. on the 4th day, the Lop-induced constipation group was orally administered a single dose of 50 mg/kg body weight Pt (Lop+Vehicle-treated group, *n* = 10) or the same volume of 1 × PBS (Lop+Pt-treated group, *n* = 10). At 9 a.m. on the 5th day, total amounts of stools, urine, water, and food in the metabolic cage of each rat were measured, and the levels of these parameters were measured as described in a previous study. After that, all SD rats were euthanized using CO_2_ gas followed by measurement of body weights, after which tissue samples of mid-colons were collected and stored at −70 °C in Eppendorf tubes until assay.

### 4.4. Measurement of Food Intake and Water Consumption

The total amounts of food and water were measured in the subset group treated with Vehicle or Pt at 9 a.m. on the 5th day using an electrical balance and a cylinder. The average values of food intake and water consumption were calculated as the difference between the amount supplied and the amount remaining.

### 4.5. Measurement of Stool Parameters and Urine Volume

Stools and urine samples were collected from SD rats without any contamination using metabolic cages (Daejong Instrument Industry Co., LTD, Seoul, Korea) throughout the experimental period. After the collection of stools excreted from each SD rat at 9 a.m. on the 5th day, their morphological images were taken using a digital camera. The total number and weight of stools were measured in duplicate per SD rat of each subgroup. These stools were dried at 60 °C for 24 h, and their weights were sequentially measured. The water content of stool was also calculated as follows:Stool water content = (A − B)/A × 100
where A is the weight of fresh stools, and B is the weight of dried stools.

In addition, the volume of urine collected at the same time as stool was measured using a cylinder.

### 4.6. Measurement of Gastrointestinal Transit Ratio and Intestinal Length

The gastrointestinal transit ratio and intestinal length were measured as described in a previous study [47]. After fasting for 18 h (except for drinking water ad libitum) prior to the experiment, SD rats were orally administered a charcoal meal (2 mL, 3% suspension of activated charcoal in 0.5% aqueous methylcellulose) (Sigma-Aldrich Co., St. Louis, MO, USA). Thirty minutes after administering the charcoal meal, SD rats were euthanized using CO_2_, and the intestinal tract from stomach to anus was collected from the abdominal cavity. Intestinal charcoal transit ratio was calculated as follows:Charcoal transit ratio (%) = [(total intestine length − transit distance of charcoal meal)/total intestine length)] × 100

In addition, the total intestinal length was measured in duplicate after taking a picture.

### 4.7. Western Blotting Analysis

Total protein homogenates were isolated from the harvested mid-colons of SD rats treated with Lop or Pt using the Pro-Prep Protein Extraction Solution (Intron Biotechnology Inc., Seongnam, Korea) according to the recommended method. After the centrifugation at 13000 rpm at 4 °C for 5 min, the concentrations of proteins were measured using a SMARTTM Bicinchoninic Acid Protein assay kit (Thermo Fisher Scientific Inc., Waltham, MA, USA). Total proteins (30 μg) were separated on 4–20% sodium dodecyl sulfate–polyacrylamide gel electrophoresis (SDS-PAGE) for 3 h and transferred onto nitrocellulose membranes for 2 h at 40 V. Each membrane was incubated with the following primary antibodies overnight at 4 °C: anti-Gα (ab128900, 1:1000, Abcam, Cambridge, UK), anti-PKC (2058s, 1:1000, Cell Signaling Technology Inc., Danvers, MA, USA.), anti-p-PKC (9376s, 1:1000, Cell Signaling Technology Inc.), anti-PI-3K (4292s, 1:1000, Cell Signaling Technology Inc.), anti-p-PI3K (4228s, 1:1000, Cell Signaling Technology Inc.), anti-MLC (ab92721, 1:1000, Abcam, Cambridge, UK), anti-p-MLC (3671s, 1:1000, Cell Signaling Technology Inc.), or anti-actin (4967s, 1:3000, Sigma-Aldrich Co.). After washing three times, each membrane was further incubated with 1:1000 diluted horseradish peroxidase-conjugated goat anti-rabbit IgG (Zymed Laboratories Inc., South San Francisco, CA, USA) for 2 h. Signal images of each protein band on the membrane were developed using a Chemiluminescence Reagent Plus kit (Pfizer Inc., Gladstone, NJ, USA) and detected using a digital camera (1.92 MP resolution) of the FluorChem^®^ FC2 Imaging System (Alpha Innotech Corporation, San Leandro, CA, USA). Finally, the density of each band in the digital image was semi-quantified using the AlphaView Program, version 3.2.2 (Cell Biosciences Inc., Santa Clara, CA, USA).

### 4.8. Quantitative Real-Time Polymerase Chain Reaction (qRT-PCR) Analysis

RT-qPCR was applied to quantify the relative levels of MUC2, AQP3, and AQP8 mRNAs. After the isolation of total RNAs from mid-colon tissues using RNA Bee solution (Tet-Test Inc., Friendswood, TX, USA), cDNA was synthesized using reverse transcriptase (Superscript II, Thermo Fisher Scientific Inc.). The relative levels of three genes were quantified by the method described in a previous study using 2 × Power SYBR Green (Toyobo Co., Osaka, Japan) [48]. The primer sequences for the above analyses were as follows: MUC2, sense primer 5′-GCTGC TCATT GAGAA GAACG ATGC-3′, antisense primer 5′-CTCTC CAGGT ACACC ATGTT ACCAG G-3′; AQP3, sense primer 5′-GGTGG TCCTG GTCAT TGGAA-3′, antisense primer 5′-AGTCA CGGGC AGGGT TGA-3′; AQP8, sense primer 5′-CTGAG GCCCT CCCAC ATCT-3′, antisense primer 5′-GGAAA GGAAC AAGGC CAACA-3′; β-actin, sense primer 5′-ACGGC CAGGT CATCA CTATT G-3′, antisense primer 5′-CAAGA AGGAA GGCTG GAAAA GA-3′. The thermal cycling conditions consisted of the holding stage (1 min at 95 °C), cycling stage (40 cycles of 15 s at 95 °C, 15 s at 57 °C, and 45 s at 72 °C), and melt curve stage (15 s at 95 °C and 60 s at 60 °C). Additional analyses, including the fluorescence intensity, the threshold values, the threshold cycle (Ct), and the housekeeping gene, were conducted as described in a previous study [49].

### 4.9. Histopathological Analysis

After the fixation of mid-colons of SD rats in 10% formalin for 48 h, these samples were subsequently embedded in paraffin wax, after which they were cut into 4 μm thick sections on slide glass. Samples were stained with hematoxylin and eosin (H&E, Sigma-Aldrich Co.) and then analyzed by light microscopy. The histopathological alterations of the mid-colon were analyzed using the Leica Application Suite (Leica Microsystems, Glattbrugg, Switzerland).

Mucin staining on the 4 μm thick tissue sections was performed using an Alcian Blue Stain kit (IHC WORLD, Woodstock, MD, USA). The morphological features in the mucin-stained mid-colon sections were observed by light microscopy.

### 4.10. Transmission Electron Microscopy (TEM) Analysis

Mid-colon tissues harvested from 8–10 rats in subset groups were fixed in 2.5% glutaraldehyde solution and subsequently dehydrated with ascending concentrations of EtOH solution. After post-fixation in 1% osmium tetroxide (OsO_4_) for 1–2 h, the tissue sample was embedded in Epon 812 media (Polysciences, Hirschberg an der Bergstrasse, Germany). Ultra-thin tissue sections (70 nm thick) were placed on a holey formvar-carbon-coated grid, after which they were stained with uranyl acetate and lead citrate. Finally, morphology of mid-colon tissues was observed by TEM (Hitachi, Tokyo, Japan).

### 4.11. Determination of GI Hormone Concentrations

The concentrations of three GI hormones, namely, cholecystokinin (CCK), gastrin, and somatostatin (SS), were determined using ELISA kits (Cusabio Biotech Co., Ltd., Wuhan, China) based on the manufacturer’s instructions. After the halogenation of mid-colon tissues (100 mg) in ice-cold 1 × PBS (pH 7.2–7.4) using a glass homogenizer (Sigma-Aldrich Co.), the homogenate of tissue was harvested for analysis. Each specific hormone antibody was mixed with tissue homogenate for 1 h at 37 °C, to which HRP–streptavidin solution was subsequently mixed for 1 h at 37 °C. Subsequently, TMP One-Step Substrate Reagent was added to the mixture, and this reaction was terminated with the stop solution. Finally, absorbance of the reaction mixture was determined at 450 nm using the Molecular Devices VersaMax Plate Reader (Molecular Devices).

### 4.12. Analysis of Fecal Microbiota

Fecal microbiota was analyzed by applying the methods described in a previous study [50]. After the collection of stools from metabolic cages, the total DNA of stool microbiota was purified from them using a DNeasy Power Soil Kit (Qiagen, Hilden, Germany) based on the manufacturer’s instructions.

The total amount of DNA was measured using the Quant-IT PicoGreen (Invitrogen, Carlsbad, CA, USA). Sequencing libraries were prepared based on the Illumina 16S Metagenomic Sequencing Library protocols used to amplify the V3 and V4 regions.

PCR was conducted with gDNA (2 ng) in 1 × reaction buffer, 1 nM dNTP mix, 500 nM of each universal F/R PCR primer, and 2.5 U Herculase II fusion DNA polymerase (Agilent Technologies, Santa Clara, CA, USA).

The 1st PCR cycle consisted of three steps: heat activation (3 min at 95 °C), amplification (25 cycles of 30 s at 95 °C, 30 s at 55 °C, and 30 s at 72 °C), and extension (5-min at 72 °C). The universal primer pair was as follows: V3-F: 5′-TCGTC GGCAG CGTCA GATGT GTATA AGAGA CAGCC TACGG GNGGC WGCAG-3′; V4-R: 5′- GTCTC GTGGG CTCGG AGATG TGTAT AAGAG ACAGG ACTAC HVGGG TATCT AATCC-3′. After the purification of the 1st PCR product using AMPure beads (Agencourt Bioscience, Beverly, MA, USA), final library construction containing the index was amplified using the NexteraXT Indexed Primer. The 2nd PCR was performed similarly to the 1st PCR, except for the application of 10 instead of 25 cycles. The final product was purified by AMPure beads, quantified using qPCR based on the qPCR Quantification Protocol Guide (KAPA Library Quantification kits for IlluminaSequencing platforms), and qualified using the TapeStation D1000 ScreenTape (Agilent Technologies, Waldbronn, Germany). Paired-end (2 × 300 bp) sequencing was performed using the MiSeq™ platform (Illumina, San Diego, CA, USA).

MiSeq raw data were curated using the Fastp program [51] and thereafter assigned to operational taxonomic units (OTUs) using the Cluster Database at High Identity with Tolerance (CD-HIT-OUT) [52]. Based on BLAST^+^ (v2.9.0) [53] in the reference DB (NCBI 16S Microbial), a representative sequence of each OTU was aligned. A comparative analysis of various microbial clusters was performed using QIIME (v1.9) [54], with the abundance and taxonomy information obtained from the above OTUs. The α-diversity information was obtained using the Shannon index and verified by examining the rarefaction curve and Chao1 values. The β-diversity was determined using weighted/unweighted UniFrac distance, and the flexibility was visualized via PCoA [54].

### 4.13. Statistical Analysis

Statistical significance between each group was evaluated using one-way analysis of variance (ANOVA) (SPSS for Windows, Release 10.10, Standard Version, IL, USA), followed by Tukey’s post hoc t-test for multiple comparisons. All values of each parameter are expressed as the means ± SD, and differences were considered statistically significant when the *p*-value was below 0.05.

## 5. Conclusions

The results of the present study indicate that Pt treatment induces the recovery of stool parameters, GI transit, histopathological and cytological alterations, GI hormone concentrations, and the mAChR signaling pathway in SD rats with Lop-induced constipation. Our results further suggest that the laxative effects of Pt are associated with alterations of the fecal microbiota profile of SD rats with Lop-induced constipation. Taken together, our findings indicate that Pt compounds derived from *E. cava* are potential therapeutic candidates for the treatment of constipation, although further research is required to verify the mode of action.

## Figures and Tables

**Figure 1 molecules-26-07209-f001:**
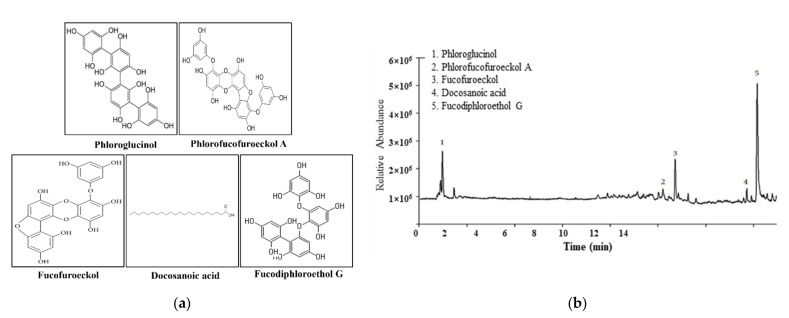
Pt chemical properties, feeding behavior, and stool parameter analyses. (**a**) Chemical structure of five identified compounds. (**b**) LC–MS analysis of Pt. Five active compounds, namely, phloroglucinol (9.35%), phlorofucofuroeckol A (3.11%), fucofuroeckol (7.55%), docasanoic acid (2.06%), and fucodiphloroethol G (31.88%), were detected as different peaks in the chromatogram. (**c**) Feeding behavior analysis. Body weight and urine volume were measured as described in Materials and Methods. The amount of water (feed) supplied and water (feed) remaining were used to measure food intake and water consumption. (**d**) Stool parameter analysis. After collection of stools from the metabolic cage, the total number and weight of stools were measured in duplicate, and their images were taken immediately using a digital camera. Stool water content was calculated as described in Materials and methods. A total of 8–10 SD rats were used for stool collection in each group, and each feeding behavior parameter and excretion parameter were assayed in duplicate. The data are reported as the mean ± SD. *, *p* < 0.05 compared to the non-treated (No) group. #, *p* < 0.05 compared to the Lop+Vehicle-treated group. Abbreviations: Lop, loperamide; Pt, phlorotannins.

**Figure 2 molecules-26-07209-f002:**
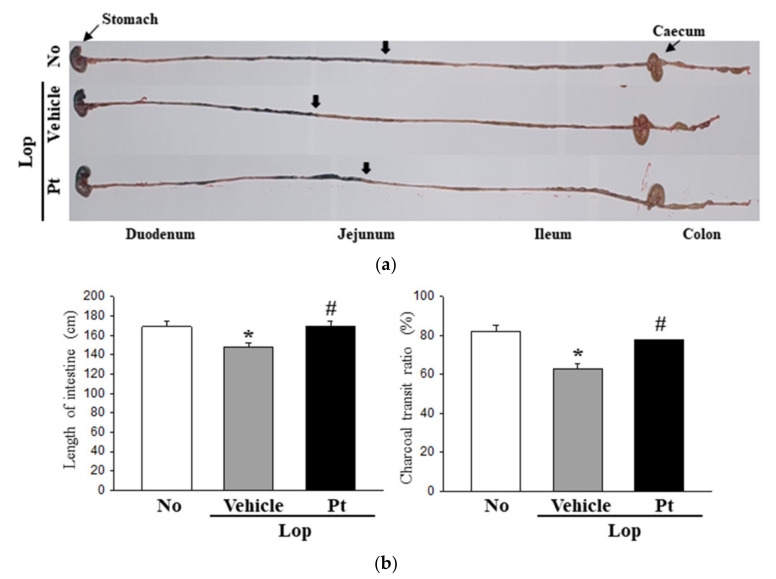
GI transit ratio and intestinal length after administration of Pt. (**a**) Actual digital image of GI. After treatment with charcoal meal powder, the total GI tract from stomach to anus was collected from each rat of subset groups, and their images were taken immediately using a digital camera. The position of charcoal meal transit is indicated with arrows. (**b**) Charcoal transit ratio and length of the intestine. The intestine length in the total GI tract was measured. The transit ratio of the charcoal meal was calculated as the ratio of the distance of the charcoal meal to the total length of the intestine. A total of 8–10 SD rats were used for the GI transit ratio analyses and measurements of intestine length in each group, and each parameter in these analyses was measured in duplicate. The data are reported as the mean ± SD. *, *p* < 0.05 compared to the non-treated (No) group. #, *p* < 0.05 compared to the Lop+Vehicle-treated group. Abbreviations: Lop, loperamide; Pt, phlorotannins.

**Figure 3 molecules-26-07209-f003:**
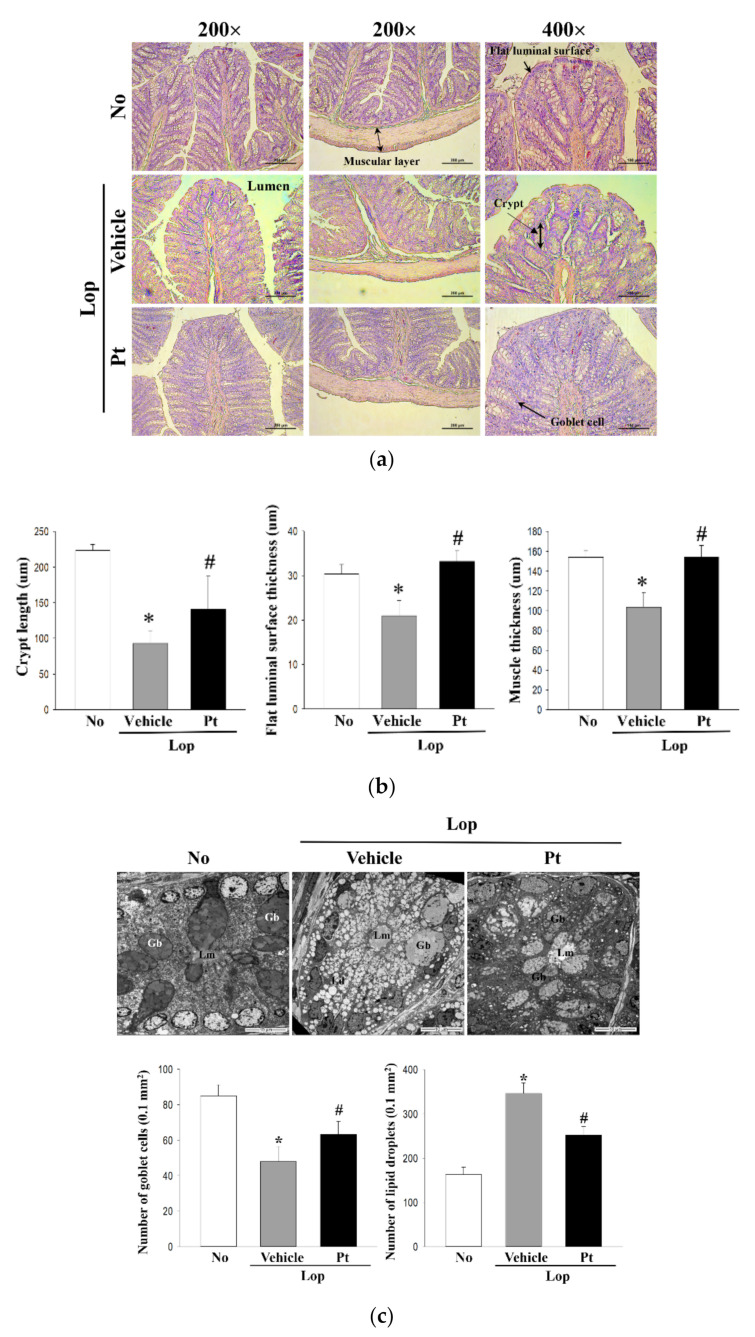
Histopathological and cytological structures of mid-colon in Lop+Pt-treated SD rats. (**a**) Histopathological images. H&E-stained sections of mid-colon from the subset groups were observed at 200 × and 400 × under a light microscope. (**b**) Histopathological parameters. Three parameters, namely, crypt length, flat luminal surface thickness, and muscle thickness, were measured using the Leica Application Suite. A total of 4–6 SD rats were used in the preparation of H&E-stained tissue slides in each group, and each parameter for histopathological properties was measured in duplicate in three different sites of each slide. (**c**) TEM image. Ultrastructure of the crypt in the mid-colon of subset groups was viewed by TEM at 1800 × magnification. The numbers of Paneth cells and lipid droplets were measured in extracellular matrix using the Leica Application Suite (Leica Microsystems, Switzerland). A total of 4–6 SD rats were used in the preparation of TEM samples, and the numbers of goblet cells and lipid droplets were measured in duplicate in each sample. Data are reported as the mean ± SD. *, *p* < 0.05 compared to the non-treated (No) group. #, *p* < 0.05 compared to the Lop+Vehicle-treated group. Abbreviations: Lop, loperamide; Pt, phlorotannins.

**Figure 4 molecules-26-07209-f004:**
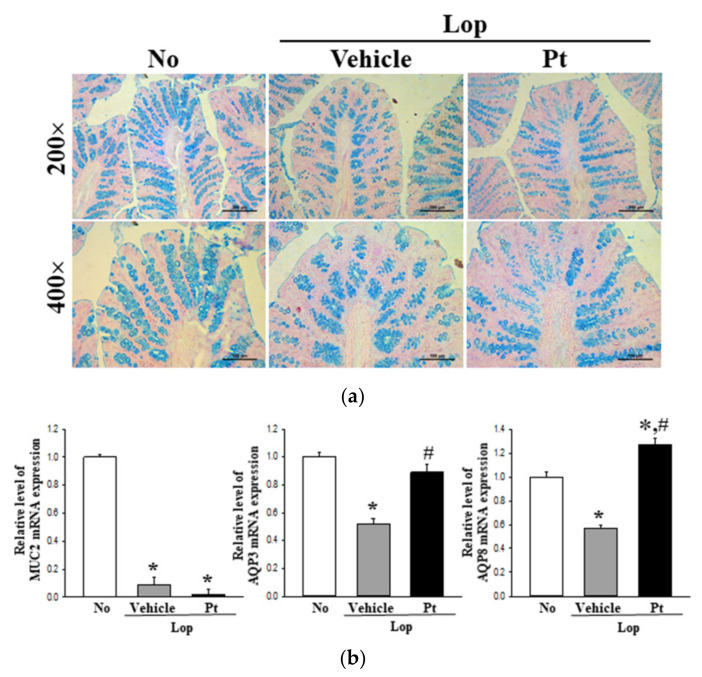
Mucin secretion and water channel expression after administration of Pt. (**a**) Mucin secretion. After staining with alcian blue, the crypt layer cells were observed at 200× and 400× magnification. A total of 4–6 SD rats were used in the preparation of alcian blue-stained tissue slides in each group, and images were taken in duplicate at three different sites of each slide. (**b**) Transcription of related genes. The transcript levels of MUC2, AQP3, and AQP8 genes were measured in the total mRNA of colons by RT-PCR using specific primers. The mRNA expression of the three genes is presented as relative values to the intensity of actin as the endogenous control protein. A total of 4–6 SD rats were used in the preparation of total RNA, and RT-PCR was performed in duplicate. Data are reported as the mean ± SD. *, *p* < 0.05 compared to the non-treated (No) group. #, *p* < 0.05 compared to the Lop+Vehicle-treated group. Abbreviations: Lop, loperamide; Pt, phlorotannins; Lm, lumen of crypt; Gb, goblet cells; Ld, lipid droplets.

**Figure 5 molecules-26-07209-f005:**
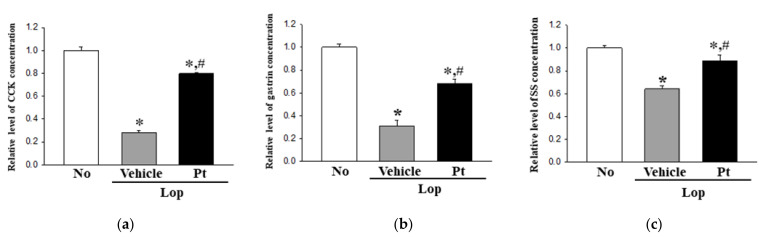
Concentrations of three GI hormones after administration of Pt. After preparation of the mid-colon homogenate, the concentrations of (**a**) CCK, (**b**) gastrin, and (**c**) SS were measured with an enzyme-linked immunosorbent assay (ELISA). The above kits for CCK, gastrin, and SS can detect up to 0.1–1000 pg/mL, 0.312–20 pg/mL, and 4.7–300 pg/mL. A total of 4–6 SD rats were used in the preparation of tissue homogenate, and hormone levels were assayed in duplicate in each sample. The data are reported as the mean ± SD. *, *p* < 0.05 compared to the non-treated (No) group. #, *p* < 0.05 compared to the Lop+Vehicle-treated group. Abbreviations: Lop, loperamide; Pt, phlorotannins; GI, gastrointestinal; CCK, cholecystokinin; SS, somatostatin.

**Figure 6 molecules-26-07209-f006:**
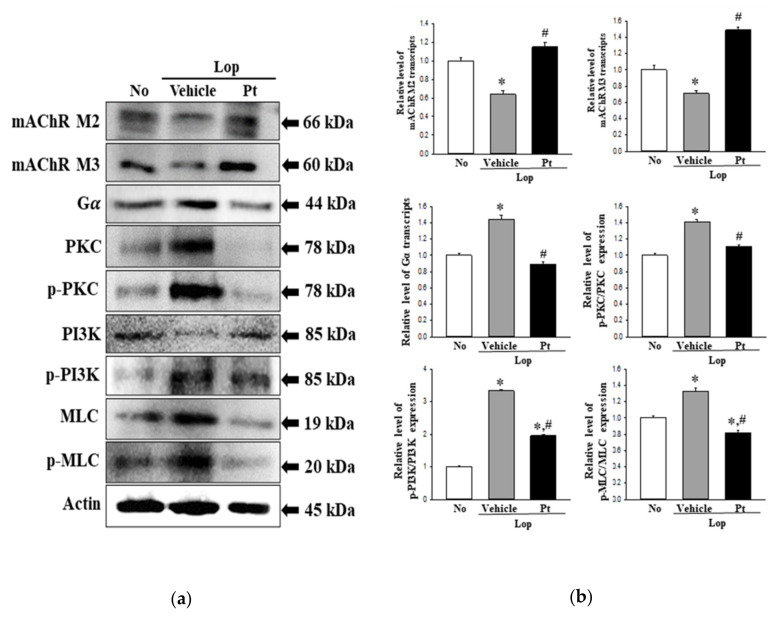
Expression of mAChRs and downstream signaling pathway. (**a**) Western blot image. (**b**) Relative levels of protein expression. After the preparation of mid-colon homogenates, the expression levels of nine proteins (mAChR M2, mAChR M3, Gα, PKC, p-PKC, PI3K, p-PI3K, MLC, and p-MLC) were determined by Western blotting. After the determination of the intensity of each band, the expression of the nine proteins was determined as relative values to the intensity of actin as the endogenous control. A total of 4–6 SD rats were used in the preparation of total homogenates, and Western blots were assayed in duplicate. The data are reported as the mean ± SD. *, *p* < 0.05 compared to the non-treated (No) group. #, *p* < 0.05 compared to the Lop+Vehicle-treated group. Abbreviations: Lop, loperamide; Pt, phlorotannins; mAChR, muscarinic acetylcholine receptor; Gα, G protein α; PKC, protein kinase C; PI3K, phosphoinositide 3-kinases; MLC, myosin light chain.

**Figure 7 molecules-26-07209-f007:**
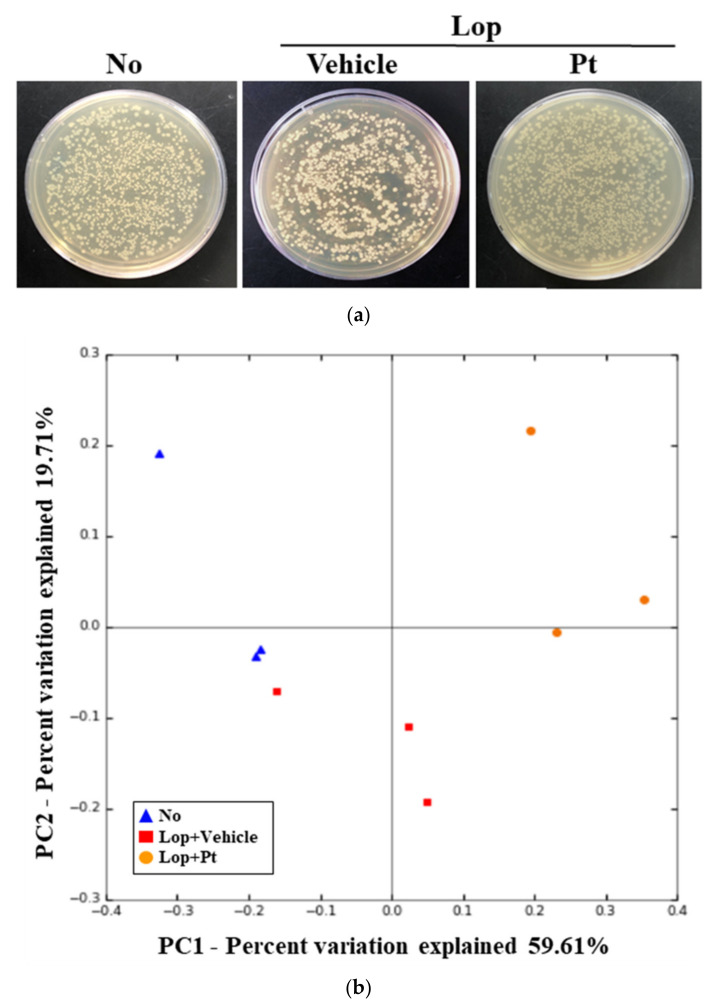
Overall analysis of fecal microbiota. (**a**) Cultivation image of fecal microbiota. After incubation for 24 h, microorganism colonies grown on nutrient agar were analyzed in a view box. (**b**) PCoA plot analysis. PCA focused on alterations of stool bacterial communities using principal components in the non-treated (No), Lop+Vehicle-treated, and Lop+Pt-treated groups. The spatial distance measure represents the degree of similarity of bacterial taxa in the stool samples. Non-treated (No): blue triangle; Lop+Vehicle: red rectangle; and Lop+Pt: orange circle. Abbreviations: Lop, loperamide; Pt, phlorotannins.

**Figure 8 molecules-26-07209-f008:**
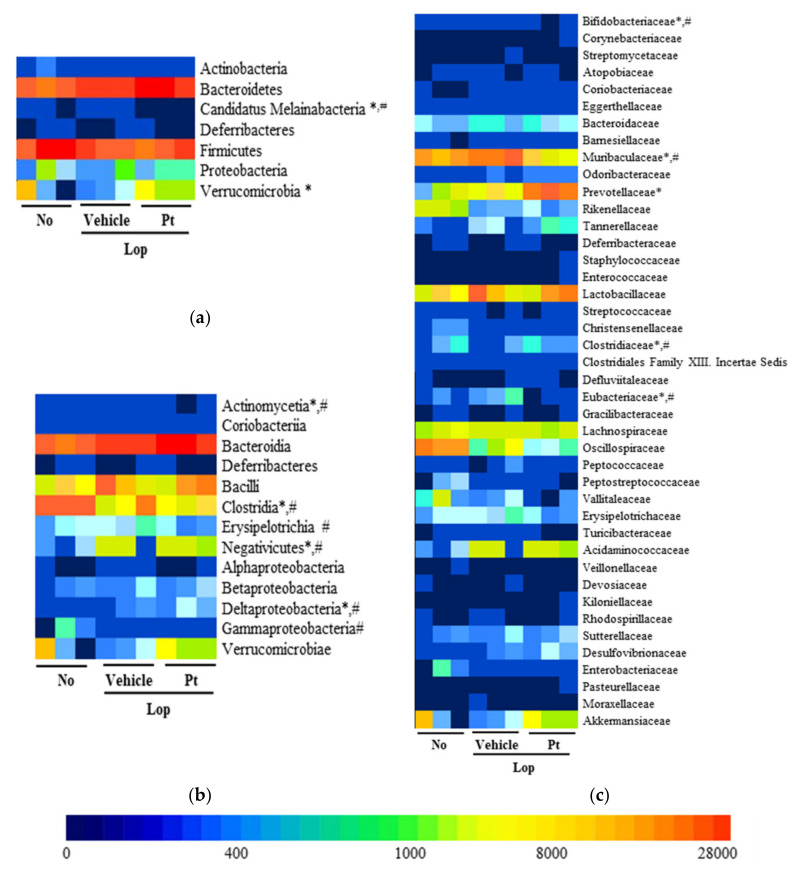
Heatmap of fecal microbiota. (**a**) Stool microbiota distribution at the phylum level. (**b**) Stool microbiota distribution at the class level. (**c**) Stool microbiota distribution at the family level. A significant difference was detected between the Lop+Vehicle- and Lop+Pt-treated groups at the bacteria phylum, class, and family levels. Different colors indicate the relative abundance of each phylum, class, and family. The data are reported as the mean ± SD. *, *p* < 0.05 compared to the non-treated (No) group. #, *p* < 0.05 compared to the Lop+Vehicle-treated group. Abbreviations: Lop, loperamide; Pt, phlorotannins.

## Data Availability

Not applicable.

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
