# Peer review of "Laxative Effects of Phlorotannins Derived from Ecklonia cava on Loperamide-Induced Constipation in SD Rats"

_molecules, 2021, doi:10.3390/molecules26237209_

Round 1

Reviewer 1 Report

In this study, the authors investigated the laxative effects of phlorotannins (Pt) derived from Ecklonia cava on chronic constipation-induced by loperamide (Lop) in SD rats. The results showed Pt improves the constipation phenotypes via regulating GI hormones and gut microbita, as well as intestinal water channel expression and mAChR signaling pathway. Overall, the study was well-designed and performed, and results are well-presented.

Some minor changes are suggested.

  1. Figure 3a is a little dim, replace it with clear one.
  2. Section 4.2 and 4.3 have same section titles, correct this one.
  3. Bacterial class names in line 214, no need to be italic. Bacterial family names in the abstract should be italic format.
  4. The degree ‘℃’ in the manuscript is not consistent, such as in line 332 and line 351.

Author Response

In this study, the authors investigated the laxative effects of phlorotannins (Pt) derived from Ecklonia cava on chronic constipation-induced by loperamide (Lop) in SD rats. The results showed Pt improves the constipation phenotypes via regulating GI hormones and gut microbita, as well as intestinal water channel expression and mAChR signaling pathway. Overall, the study was well-designed and performed, and results are well-presented.

Some minor changes are suggested.

  1. Figure 3a is a little dim, replace it with clear one.

☞ That has been corrected.

  1. Section 4.2 and 4.3 have same section titles, correct this one.

☞ That has been corrected.

  1. Bacterial class names in line 214, no need to be italic. Bacterial family names in the abstract should be italic format.

☞ That has been corrected.

  1. The degree ‘℃’ in the manuscript is not consistent, such as in line 332 and line 351.

☞ That has been corrected.

Reviewer 2 Report

The study of Kim et al is of interest. Yet, in my opinion, to be considered for publication in Molecules journal, the authors must improve the data/discussion with regard to the chemical characterization of the extract that they consider to be rich in Pt.

Additional specific questions:

  • What are the levels of Phlorotannins in the extract?
  • Fig 1a) and b) must be cited in the text before their appearance
  • Relative abundance (Fig 1b) should be in percentage
  • Please detail the elution LC-MS program
  • Please revise title of section 4.3

Author Response

The study of Kim et al is of interest. Yet, in my opinion, to be considered for publication in Molecules journal, the authors must improve the data/discussion with regard to the chemical characterization of the extract that they consider to be rich in Pt.

☞ According to your comments, we have further discussed above issue in Discussion part as following;

“Meanwhile, Pt are produced as high hydrophilic components through the polymerization of phloroglucinol (1,3,5-trihydroxybenzene) monomer and the acetate–malonate pathway [41]. It has been classified into four subclasses, fuhalols and phlorethols, fucols, fucophloroethols and eckols [42]. Especially, some subclasses of Pt were isolated and characterized from E. cava. They include phloroglucinol [43], eckol [43,44], fucodiphloroethol G [43], phlorofucofuroeckol A [43,44], 7-Phloroeckol [43], dieckol [43] and 6,6’-Bieckol [43,45] although they have been reported in different papers. In the present study, LC-MS analyses were identified only five compounds including phloroglucinol, phlorofucofuroeckol A, fucofuroeckol, and fucodiphloroethol G. This difference is thought to be due to the extraction method and the quality of the raw material used.”

Additional specific questions:

What are the levels of Phlorotannins in the extract?

☞ That has been further described in Results part.

Fig 1a) and b) must be cited in the text before their appearance.

☞ They have been cited in Results part.

Relative abundance (Fig 1b) should be in percentage.

☞ That has been corrected in legend of Figure 1.

Please detail the elution LC-MS program.

☞ That has been corrected in Materials and Methods part.

Please revise title of section 4.3

☞ That has been corrected. 

Reviewer 3 Report

The paper indicates a new possibility to a better constipation manage. The conducted study is interesting and the obtained by Authors results are promising. 

Author Response

The paper indicates a new possibility to a better constipation manage. The conducted study is interesting and the obtained by Authors results are promising. 

☞ Thank you very much.

Reviewer 4 Report

The paper submitted by Kim et al presents a high quality research on the bioactivity of Ecklonia cava in animal model. Several aspects including the investigation of the influence of the extract on gut microbiota is summarized. The paper should be published in Molecules after addressing some comments given below:

1) because LC-MS analysis was performed please provide MS data for compounds present in the extract used for bioassays (please provide table with retention times, MS1 and MS2 data); chromatogram shown in the text should be bigger and chemical structures of all detected and identified compounds should be depicted

2) in fig 8 - please provide easy to read info on changes in the chemical composition of microbiota before and after the treatment - please show only families, phyla and genera which changed abundance as a result of experiments

Author Response

The paper submitted by Kim et al presents a high quality research on the bioactivity of Ecklonia cava in animal model. Several aspects including the investigation of the influence of the extract on gut microbiota is summarized. The paper should be published in Molecules after addressing some comments given below:

  • because LC-MS analysis was performed please provide MS data for compounds present in the extract used for bioassays (please provide table with retention times, MS1 and MS2 data); chromatogram shown in the text should be bigger and chemical structures of all detected and identified compounds should be depicted

☞ According to your comments, chromatogram has been replaced with bigger one in Figure 1b and chemical structure has been inserted in Figure 1a.

  • in fig 8 - please provide easy to read info on changes in the chemical composition of microbiota before and after the treatment - please show only families, phyla and genera which changed abundance as a result of experiments

☞ This is good point. But, in order to help reader understand, it is necessary to present the changed group and the unchanged group at the same time. Under this condition, significant changes are indicated by an asterisk. Please understand this situation.

Round 2

Reviewer 2 Report

The authors have improved the manuscript according to my comments.